# The Mitochondrial tRNA^Ser(UCN)^ Gene: A Novel m.7484A>G Mutation Associated with Mitochondrial Encephalomyopathy and Literature Review

**DOI:** 10.3390/life13020554

**Published:** 2023-02-16

**Authors:** Eugenia Borgione, Mariangela Lo Giudice, Sandro Santa Paola, Marika Giuliano, Francesco Domenico Di Blasi, Vincenzo Di Stefano, Antonino Lupica, Filippo Brighina, Rosa Pettinato, Corrado Romano, Carmela Scuderi

**Affiliations:** 1Unit of Neuromuscular Diseases, Oasi Research Institute-IRCCS, Via Conte Ruggero 73, 94018 Troina, Italy; 2Unit of Psychology, Oasi Research Institute-IRCCS, Via Conte Ruggero 73, 94018 Troina, Italy; 3Unit of Neurophysiopathology, Department of Biomedicine, Neuroscience, and Advanced Diagnostics (BiND), University of Palermo, 90129 Palermo, Italy; 4Unit of Pediatrics and Medical Genetics, Oasi Research Institute-IRCCS, Via Conte Ruggero 73, 94018 Troina, Italy; 5Research Unit of Rare Diseases and Neurodevelopmental Disorders, Oasi Research Institute-IRCCS, Via Conte Ruggero 73, 94018 Troina, Italy; 6Medical Genetics, Section of Medical Biochemistry, Department of Biomedical and Biotechnological Sciences, University of Catania, 95123 Catania, Italy

**Keywords:** mitochondrial DNA, tRNA^Ser(UCN)^, homoplasmic mutation, encephalomyopathy

## Abstract

Mitochondrial tRNA^Ser(UCN)^ is considered a hot-spot for non-syndromic and aminoglycoside-induced hearing loss. However, many patients have been described with more extensive neurological diseases, mainly including epilepsy, myoclonus, ataxia, and myopathy. We describe a novel homoplasmic m.7484A>G mutation in the tRNA^Ser(UCN)^ gene affecting the third base of the anticodon triplet in a girl with profound intellectual disability, spastic tetraplegia, sensorineural hearing loss, a clinical history of epilepsia partialis continua and vomiting, typical of MELAS syndrome, leading to a myoclonic epilepticus status, and myopathy with severe COX deficiency at muscle biopsy. The mutation was also found in the homoplasmic condition in the mother who presented with mild cognitive deficit, cerebellar ataxia, myoclonic epilepsy, sensorineural hearing loss and myopathy with COX deficient ragged-red fibers consistent with MERRF syndrome. This is the first anticodon mutation in the tRNA^Ser(UCN)^ and the second homoplasmic mutation in the anticodon triplet reported to date.

## 1. Introduction

Mitochondria are essential cytoplasmic organelles in eukaryotic cells that generates about 90% of cell energy through the aerobic production of adenosine triphosphate (ATP) by the oxidative phosphorylation (OXPHOS) process in mammalian cells; this process depends on five intramembrane complexes and coenzyme Q10 and citocrome c which carry on electrons among them [1].

Mitochondria, also, play an important role in many processes, including β-oxidation of fatty acids and the tricarboxylic acid cycle (TCA), calcium handling, in regulating apoptosis and participating in the cell cycle [1].

Mitochondria have an own DNA comprises 16,569 base pairs in multiple copies. The process of replication and transcription of mitochondrial DNA (mtDNA) and protein translation occur independently. However, most mitochondrial proteins are encoded by nuclear DNA (nDNA); in fact, only 13 of the mitochondrial respiratory chain (RC) complex proteins are encoded by mtDNA as well as all 22 mitochondrial transfer RNAs (mt-tRNAs), and 2 ribosomal RNA (mt-rRNA).

Defects in either mtDNA or nDNA may result in mitochondrial dysfunction leading to diseases.

Mitochondrial disorders are more common than previously thought affecting about 1 out 5000 adults in Europe [2]. They might be inherited or sporadic disorders; indeed, many patients with mitochondrial disease do not have a relevant medical history. Because brain and muscle have high-energy requirements, they are vulnerable to the dysfunction of oxidative ATP production. As a result, although mitochondria are ubiquitous and every tissue can be theoretically affected, mitochondrial disorders frequently manifest as encephalomyopathies, that are multisystemic disorders with prominent involvement of the central nervous system and the skeletal and cardiac muscles. 

The primary molecular defects in these diseases are mutations (deletions, duplications, point mutations) in mtDNA, which produce defects in one or more complexes of the mitochondrial RC.

Since the discovery of the first mtDNA variation in 1988, over 400 point mutations in the mitochondrial genome have been associated with human diseases. It is interesting that only ~8% of the entire mitochondrial genome is represented of mt-tRNA genes; however, a large number of point mutations occurs in mt-tRNA genes and the frequency of pathogenic variants, that impair mtDNA translation thus leading to neurological syndromes, is significantly higher than that of mitochondrial messenger RNA [3] according to the human mitochondrial genome database MITOMAP.

Many syndromes have been connected with specific mutations and some phenotypes can guide an easy recognition in typical patients. On the other hand, sometimes there is a high variability within the same family with different phenotypes in people carrying the same mutation. Leber hereditary optic neuropathy (LHON) is the most common mtDNA-related disorder in adults, with subacute blindness in young males due to bilateral optic atrophy. Approximately 95% of LHON cases are due to three primary mutations (m.11778G>A, m.3460G>A, m.14484T>C) affecting genes encoding complex I subunits [2]. Mitochondrial encephalomyopathy, lactic acidosis and stroke-like episodes (MELAS), caused in the most cases by A-to-G transition at nucleotide 3243 in tRNA^Leu(UUR)^ gene [4], usually presents in children and young adults after normal milestones with headache, hemiparesis, cortical blindness, and hemianopia due to infarct not corresponding to the distribution of major vessels. Myoclonus, epilepsy with ragged red fibers (MERRF), caused by A-to-G transition at nucleotide 8344 in tRNA^Lys^ gene [5], is characterized by myopathy, seizures, myoclonus, and ataxia. Maternally inherited myopathy and cardiomyopathy (MMC), caused by A-to-G transition at nucleotide 3260 in tRNA^Leu(UUR)^ affect young adults [6]. Finally, maternally inherited Leigh syndrome (MILS), a severe infantile encephalopathy with symmetric lesions in the basal ganglia, and neuropathy, ataxia, and retinitis pigmentosa (NARP) are associated with the m. 8993T>G mutation in *ATP6*. It is interesting that this mutation is expressed as NARP when mutant mtDNA proportion is 70–90% of total mtDNA and as MILS when this proportion is >90% [2]. Also, a few families have been described with distinct mitochondrial syndromes due to mutations in the mitochondrial tRNA^Ser(UCN)^ [7].

Mutations in mtDNA are transmitted by maternal inheritance because at fertilization all mitochondria are derived from the oocyte [1]. Hence, a mother carrying a mtDNA mutation will transmit it to all offspring, both males and females, but only her daughters will pass it to the progenies. 

When a mtDNA pathogenic mutation affects some but not all genomes in a cell or in a tissue, the whole individual will harbor two population of mtDNA, the one normal and the other mutant, a condition known as heteroplasmy [1]. Mutations in mtDNA are usually heteroplasmic with mutant and wild type mtDNAs coexisting in tissues. It is assumed that in normal tissues all mtDNA were considered identical (homoplasmy), but it is not correct at all. Indeed, NGS techniques have revealed the coexistence of mutated mtDNA variants (among 0.2 and 2% of heteroplasmy) in unaffected individuals. Definitively, the concept of heteroplasmy is not absolute, but a minimal critical number of mutant genomes in affected tissue is needed for biochemical and clinical manifestations (threshold effect). As a consequence, even small decreases in the amount of wild type mtDNA may be sufficient to cause disease in such conditions. This might account for the high-variable clinical phenotypes involving many organs and tissues even in the same family. Furthermore, there is a possible variability in mtDNA even in the same subject at different times this may lead to phenotype progression with age and disparities among different tissues.

In this context, it is really difficult to determine the pathogenicity of novel mutations in mt-tRNA genes, particularly when are associated with dominance, segregation in a tissue or homoplasmy. However, in some conditions homoplasmic mutations have been demonstrated to have a pathogenic role.

This clinical evidence supports the idea that other, yet unidentified, factors apart from heteroplasmy (i.e., polyplasmy or mitotic segregation) are determinant for the resultant phenotype in the presence of homoplasmic mutations. Further studies are needed to clarify the role of homoplasmic mutations in mitochondrial disorders.

We report a novel m.7484A>G homoplasmic mutation in the tRNA^Ser(UCN)^ gene affecting the third base of the anticodon triplet in a girl and her mother presenting with mitochondrial encephalomyopathy and sensorineural hearing loss.

## 2. Case Report

The proband was a seven-year-old girl, born full term by Caesarean delivery. During pregnancy, the mother presented with epileptic seizures and was treated with antiepileptic drugs. The patient did not have perinatal asphyxia. Head control was reached at the age of three months and sitting without support at the age of eight months.

At seven months, muscular hypotonia and fever-related subcontinuous seizures were observed, characterized by eyes revulsion and head jerks. At the age of nine months, she presented a seizure characterized by eyes revulsion and an extension of the arms, followed by generalized tonic-clonic jerks and skin cyanosis, which was stopped one hour later through the intravenous administration of diazepam. In the following days, vomiting episodes succeeded. At ten months, after a vomiting episode, she presented a seizure characterized by continuous jerks in her left limbs, lasting two hours, and followed by prolonged Todd paralysis. Corticosteroid and diazepam were administered. After fifteen days, subcontinuous myoclonia appeared, involving either the right arm or leg, sometimes migrating on the other side of the body. Continuous paroxysmal activity prevalent on the right cerebral hemisphere—configuring a picture of epilepsia partialis continua—was described on EEG recordings. The patient was treated with numerous antiepileptic drugs (carbamazepine, valproic acid, diazepam, clobazam, phenobarbital and lamotrigine) and ACTH, but the seizures persisted. At that time, she progressively lost the psychomotor acquisitions and developed a severe spastic tetraplegia; her health worsened, showing respiratory problems and spontaneous fractures. Sensorineural hearing loss was found.

Serial brain imaging (MRI at ten months, CT at eleven months and MRI at five years, respectively), documented progressive severe brain atrophy with less involvement of the posterior cranial fossa (Figure 1A–D).

At the age of seven years, on clinical examination, facial dysmorphisms were found. Neurological evaluation revealed sopor, bilateral ptosis, diffuse muscular hypotrophy and hypertonia, diffuse brisk tendon reflexes, bilateral Babinski sign, flexor asymmetrical posture, thoracic dysmorphism, scoliosis, left hip luxation, and equinus feet; neither head control nor spontaneous movements of her arms were seen. Although the patient was taking phenobarbital, diazepam, valproic acid, and oxacarzepine, she presented subcontinous myoclonia involving both the right and left arms and face, configuring a mioclonic epilepticus status.

EEG studies showed nearly continuous paroxysmal activity with numerous burst suppressions, diffuse in both or only one side of the cerebral hemispheres, particularly the right one. The ECG and echocardiogram showed right ventricular hypertrophy. Brainstem auditory evoked potentials (BAEPs) did not record any bilateral response. Fundoscopic examination showed bilateral papillar atrophy. On laboratory investigations, the blood lactate and pyruvate were normal.

The patient died after two years of our examination for epilepticus status.

Since the age of 18 years, the mother, 34 years old, presented with muscular weakness, generalized tonic-clonic seizures and myoclonic jerks in her hands. Then, progressive walking difficulties appeared. A neurological examination showed mild cognitive deficit, mild diffuse muscular weakness, mild muscular hypotrophy, which was more evident in her legs, and cerebellar signs including dysarthria, dysmetria, intentional tremor and ataxic gait. Her ocular fundus examination and visual evocated potentials were normal. BAEPs documented sensorineural hearing loss. EEG studies showed multifocal spikes and waves. Electromyography showed myogenic signs. A brain MRI revealed moderate brain atrophy. In laboratory investigations, her blood lactate and pyruvate were normal.

It was reported that the proband’s maternal grandmother suffered from epilepsy and ataxia; her maternal uncle was deaf; a brother of her maternal grandmother, who was blind, died in advanced old age. No members of this family were available for clinical and genetic examination.

Informed consent for study participation was received. The study was carried out in accordance with the Declaration of Helsinki of 1964 and its later amendments, and the Ethics Committee of the Oasi Research Institute-IRCCS, Troina (Italy), approved the protocol on 5 April 2022 (2022/04/05/CE-IRCCS-OASI/52).

## 3. Materials and Methods

### 3.1. Morphologic and Biochemical Analysis

Muscle specimens of the patient and her mother were obtained from the right biceps under local anesthesia and immediately frozen in liquid nitrogen-cooled isopentane.

For histological analysis, 8 µm-thick cryostatic sections were picked and stained for routine process with hematoxylin and eosin (H&E), Gomori’s trichrome, oil red O, periodic acid-Shiff, NADH tetrazolium reductase, succinate dehydrogenase (SDH), cytochrome c oxidase (COX) and myofibrillar adenosine triphosphatase (ATPase) at pH 4.3, 4.7 and 9.4 as described [8].

Mitochondrial enzyme activities on total muscle were determined using spectrophotometric assays in supernatant obtained homogenizing in 9 volumes of 0.15 M KC1, 50 mM Tris-HC1, pH 7.4 and after centrifugation at 750 g for 15 minutes, as described [9]. The activities of the RC complexes were referred to that of citrate synthase (CS) to correct for mitochondrial volume.

### 3.2. Molecular Genetic Analysis

The total mtDNA isolated from the frozen muscle of the patient and the mother was tested for mtDNA large-scale rearrangements by Southern blot and long-PCR and screened for sequence variants by Sanger.

The entire mtDNA was amplified in 24 partially overlapping PCR fragments, as described elsewhere [10]. The derived PCR fragments were subjected to direct sequencing with the Big Dye Terminator Cycle Sequencing Kit and an ABI PRISM 310 Genetic Analyzer (Perkin Elmer, Waltham, MA, USA).

To confirm the mutation at nt 7484 and to quantify the amount of mutant mtDNAs in the blood and muscle of the patient and her mother, we amplified a 147 bp fragment incorporating a mismatch in the forward primer and performed RFLP analysis using Xho I.

The sequence of the PCR primers used were: Forward 5′-AACCCCCCAAAGCTGGTCTC-3′ and Reverse 5′-GACCTACTTGCGCTGCATG-3′. To amplify the target sequence, we used the following PCR conditions: after an initial 5 min denaturation step at 94 °C, 1 min denaturation at 94 °C, 1 min annealing at 56 °C and 45 s extension 72 °C for 30 cycles and a final 5 min elongation step at 72 °C. To quantitate the proportions of mutant and wild-type mtDNAs, approximately 0.5 µCi of [α-32P]-d CTP was added immediately before the last cycle. The T to C mismatch at position 7480 in the forward primer creates a restriction site for the enzyme Xho I in the mutant allele. The 147 bp PCR product from the mutant allele is cut into fragments of 129 and 18 bp; the wild-type allele remains uncut.

The digested products were separated on a 12% non-denaturing polyacrylamide gel and subjected to autoradiography.

The measure of mtDNA copy number was performed on muscle biopsy by Quantitative Real Time PCR based on Taqman fluorescence. For each sample qRT-PCR reactions were carried out in duplicate. RT products were amplified under the thermal cycling conditions: one cycle for 2 min at 50 °C, one cycle of 15 min at 95 °C and 40 cycles for 15 s at 94 °C followed by 1 min at 60 °C. The data obtained were quantified using the comparative DDCt method. In this analysis, the amount of mtDNA was compared with the amount of the nuclear gene cluster encoding the 18S rRNA on chromosome 21, contained in the same sample. The mtDNA/18S rRNA ratio obtained in the tested sample was expressed as a percentage of the mean value obtained in control samples, which represent the 100% value.

To exclude mutations in the nuclear genes, the genomic DNA of the patient and her mother was extracted from the peripheral blood using standard protocols and Whole exome sequencing (WES) was performed in both. Nextera Flex for Enrichment Sample Prep kit (Illumina, San Diego, CA, USA) was used according to the manufacturer’s instructions. The libraries were sequenced in an Illumina NextSeq500 using a 2 × 150 paired-end reads protocol. The reads were aligned to hg19, and the variants were identified through the GATK pipeline. In accordance with the pedigree and phenotype, priority was given to rare variants [<1% in public databases, including 1000 Genomes project, NHLBI Exome Variant Server, and Exome Aggregation Consortium (ExAC v0.2)]. Deleterious single-nucleotide variants (SNVs) were predicted by SIFT (http://sift.bii.a-star.edu.sg/ (accessed on 15 January 2022)), PolyPhen-2 (http://genetics.bwh.harvard.edu/pph2/ (accessed on 15 January 2022)), and MutationTaster (http://www.mutationtaster.org/ (accessed on 15 January 2022)) programs. Candidate SNVs were validated by the ABI3130 sequencer.

### 3.3. Western Blot Analysis

Fifty micrograms of proteins obtained from the muscle homogenates were denatured and loaded on sodium dodecyl sulphate polyacrylamide gels. Subsequently, the proteins were transferred to a nitrocellulose membrane and subjected to western blotting. The membranes were blocked in Tris-Buffered Saline and Tween20 (TBST) (150 mM NaCl, 10 mM tris-HCl, pH 7.5 and 0.1% Tween20) containing 5% (*w/v*) milk, then incubated with the corresponding primary and secondary antibodies. The primary antibodies used for this experiment were the goat anti-ND1, mouse anti-CO1, rabbit anti-ATP6 (Santa Cruz, Santa Cruz, CA, USA) and β-actin (Cell Signaling, Danvers, MA, USA).

## 4. Results

Muscle biopsy of the proband, performed at the age of six years, and her mother showed an increase in perimysial connective, a variation in fiber size, multiple central nuclei, fat accumulation, and a diffuse and marked reduction in the COX activity with almost every fiber partially or totally depleted (Figure 2). The mother also had some fibers with the aspect of ragged red (Figure 2B).

The biochemical measurements of the RC complex activities revealed marked COX deficiency in the patient (0.77, nv 2.21–4.19; CS 16.87, nv 10.33–16.51) and her mother (0.88, nv 2.21–4.19; CS 19.94, nv 10.33–16.51). The values are expressed in nmol/min/mg.

The muscle mtDNA Southern blot and long-PCR excluded the presence of large-scale deletions. The sequence analysis of the total mtDNA revealed a novel nucleotide change: an A to G substitution at position 7484 in the tRNA^Ser(UCN)^ gene (Figure 3A).

However, as the gene is encoded on the light strand, the base change corresponds to a T to C in the tRNA^Ser(UCN)^ transcript (Figure 3B).

A PCR assay was designed to verify the presence of the mutation and to measure the proportions of mutant and wild type mtDNAs. The assay incorporates a mismatch forward primer, creating a unique Xho I restriction site in the PCR product when the mutant allele is present. PCR/RFLP analysis showed that this mutation was apparently homoplasmic in the samples tested (Figure 3C). The mutation was absent from 110 independent normal controls and 42 patients with a putative mitochondrial disorder, with or without a precise molecular diagnosis.

The mtDNA copy number analysis showed a normal mtDNA content in the mother’s muscle when compared to the tissue and age-matched pooled controls (Figure 4). Unfortunately, no biological tissue of the proband was available for this analysis.

WES ruled out pathogenic point mutations in the nuclear genes.

To determine the consequences of the m.7484A>G mutation on mitochondrial protein synthesis, western blot analysis was carried out to examine the levels of four respiratory complex subunits, with β-actin as a loading control. As shown in Figure 5, the levels of ND1, COI and ATP6 were markedly decreased in the muscle of the patient and partially in her mother compared with the control tissue.

## 5. Discussion

We report the case of a girl presenting with profound intellectual disability, spastic tetraplegia, myoclonic epilepticus status, sensorineural hearing loss and myopathy, with severe COX deficiency at muscle biopsy that carried a novel homoplasmic m.7484A>G mutation in the tRNA^Ser(UCN)^ gene.

Her clinical history was characterized by epilepsia partialis continua and vomiting, typical of MELAS syndrome [11], leading to generalized mioclonus epilepticus status and severe cerebral atrophy with rapid motor and cognitive deterioration. Her mother, carrying the same mutation, showed cognitive deficit, cerebellar ataxia, myoclonic epilepsy, sensorineural hearing loss and myopathy with COX deficient ragged-red fibers consistent with MERRF syndrome.

Mitochondrial tRNA^Ser(UCN)^ is considered a hot-spot for non-syndromic and aminoglycoside-induced hearing loss, including the m.7444G>A, m.7445A>C, m.7445A>G, m.7510T>C, m.7511T>C mutations [12,13,14,15,16]. However, many patients have been described with more extensive neurological disease including progressive external ophthalmoplegia (PEO), epilepsy, myoclonus, ataxia, and myopathy [17,18,19,20].

Moreover, mitochondrial tRNA^Ser(UCN)^ mutations were associated with non-neurological disease such as cardiovascular disease [21], hypertension [22], renal disease [23], polycystic ovary syndrome and insulin resistance [24].

Table 1 shows all of the mutations identified in the tRNA^Ser(UCN)^ and the clinical phenotypes from the literature.

The clinical phenotype of our patients is consistent with other cases described in literature, including both deafness, encephalopathy, and muscular impairment. 

A case of epilepsia partialis continua leading to intractable epilepticus status was described associated with homoplasmic m.7472insC [20], and MELAS-MERRF overlap syndrome was described associated with m.7512T>C [54].

It is also interesting that most mutations are associated with some degree of phenotypic heterogeneity. For example, the m.7472insC and m.7445A>G, that manifest primarily as hearing impairment, are also associated with a disorder affecting other tissues [29,40]; it may be due to specific effects of several mutations on processes different from the synthesis of tRNA^Ser(UCN)^ [57].

Some features support pathogenicity of the m.7484A>G mutation. 

First, the mutation was absent in 56910 mtDNA genomes according to the website Mitomap and in our 162 Italian individuals (normal and disease control subjects), indicating that it is unlikely in the general population or in association with other known pathogenic mtDNA mutations. In addition, it was associated with abnormal morphological and biochemical mitochondrial features in the proband and in her mother. Moreover, a marked decrease in level of four mtDNA-encoded polypeptides was observed using a western blot analysis, supporting a correlation of the m.7484A>G mutation with the impairment of mitochondrial protein synthesis.

This change disrupts the highly conserved third base of the anticodon triplet of tRNA^Ser(UCN)^, that may compromise its function as tRNA identity determinant. In fact, many anticondon sequences are recognized by their cognate aminoacyl tRNA synthetase for specific amino acid addition [58,59] and one single base change may lead to noncharging [60], resulting in a lack of functional tRNA^Ser(UCN)^ which is necessary for protein synthesis.

Furthermore, heteroplasmy has been traditionally considered important evidence for the pathogenicity of a mtDNA mutation, and an important determinant of the clinical phenotype [9], while homoplasmic changes generally tend to be underestimated.

However, we believe that the mutation found, although homoplasmic in the proband and her mother, is pathogenic. Indeed this is strongly supported by literature with the increasing evidence that pathogenic homoplasmic mtDNA defects are more common than previously thought [61], in particular in the tRNA^Ser(UCN)^ gene [7,12,13,14,16,19,20,33]. In fact, such mutations are often homoplasmic or at high levels of heteroplasmy, suggesting that high threshold of the mutated mtDNA must accumulate for pathogenicity [28].

Mutations affecting the anticodon triplet of tRNAs, are hypothetically likely to be pathogenic because interfere with the decoding process of a tRNA. Suppositionally these variants are incompatible with early developmental stages or lethal in embryogenesis and for this reason are rarely reported [62]; in fact, only seven previous mutations have been described associated to different phenotypes, especially encephalomyopathies [63,64,65,66,67,68,69]. This is the first mutation occurring in the anticodon of tRNA^Ser(UCN)^ and the second homoplasmic mutation in the anticodon triplet. In particular, our mutation was homoplasmic both in the proband and in the less severe affected relative, instead for the mutation reported by Zanssen et al. [68], there was no information about the degree of mutation in the relatives. Indeed, anticodon mutations already described are generally heteroplasmic because homoplasmic mutations in this location must be considered lethal. Our patient survived until the age of 9 years, while her mother is still alive, confirming the less pathogenicity of tRNA^Ser(UCN)^ gene.

Moreover, the reasonable number of mutations in this tRNA gene, reported in literature, confirms serine (UCN) as one of the most common sites for mtDNA mutations.

Finally, the possibility that this mutation might represent a neutral polymorphism, is not supported by the absence of mutations in the exome.

## Figures and Tables

**Figure 1 life-13-00554-f001:**
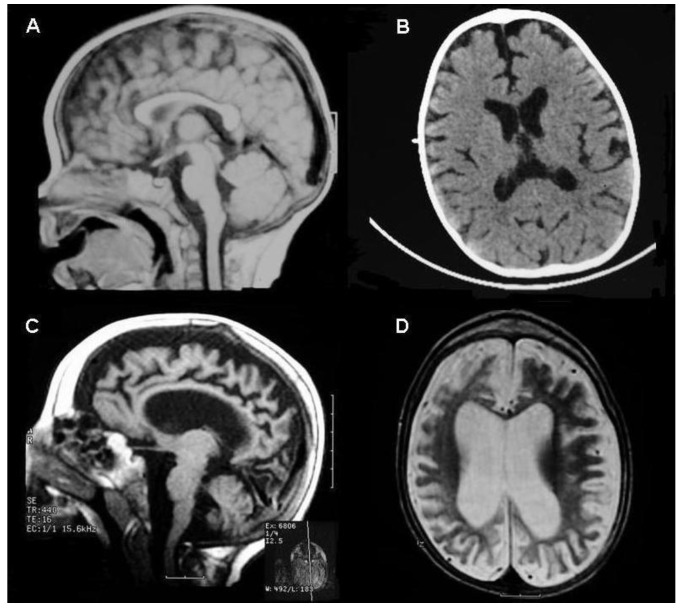
Serial brain imaging of the patient: MRI at 10 months (**A**), CT at 11 months (**B**), and MRI at 5 years (**C**,**D**) showing progressive severe brain atrophy with less involvement of posterior cranial fossa.

**Figure 2 life-13-00554-f002:**
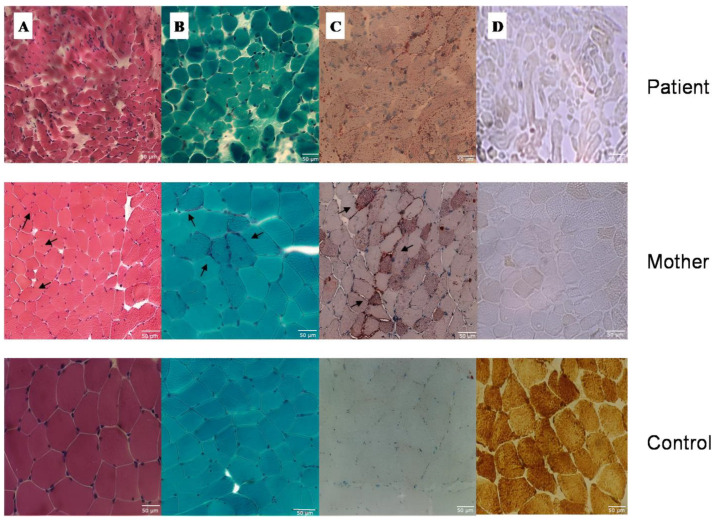
Muscle histochemistry: (**A**) Hematoxylin and eosin stain shows central nuclei (arrows). (**B**) Gomori’s trichrome stain shows RRFs (arrows). (**C**) Oil red O stain shows lipid accumulation (arrows). (**D**) Cytochrome c oxidase (COX) stain shows COX-deficient fibers.

**Figure 3 life-13-00554-f003:**
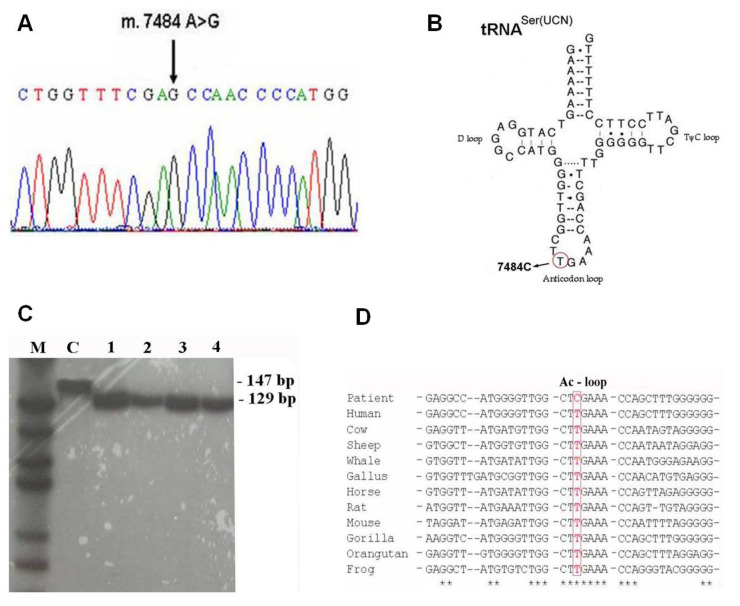
(**A**) Electropherogram of the mitochondrial DNA region encompassing the m.7484A>G mutation. (**B**) Proposed secondary structure of the tRNA^Ser(UCN)^. The gene is encoded on the light strand and the base change is shown as T to C in the tRNA. (**C**) PCR/RFLP analysis of the m.7484A>G mutation. M = molecular Marker, C = control, 1 = patient blood, 2 = patient muscle, 3 = mother blood, 4 = mother muscle. (**D**) Comparison of mitochondrial tRNA^Ser(UCN)^ among several species. Position 7484 is indicated by the boxed area. * The asterisk indicates the nitrogenous base conserved along the evolutionary scale.

**Figure 4 life-13-00554-f004:**
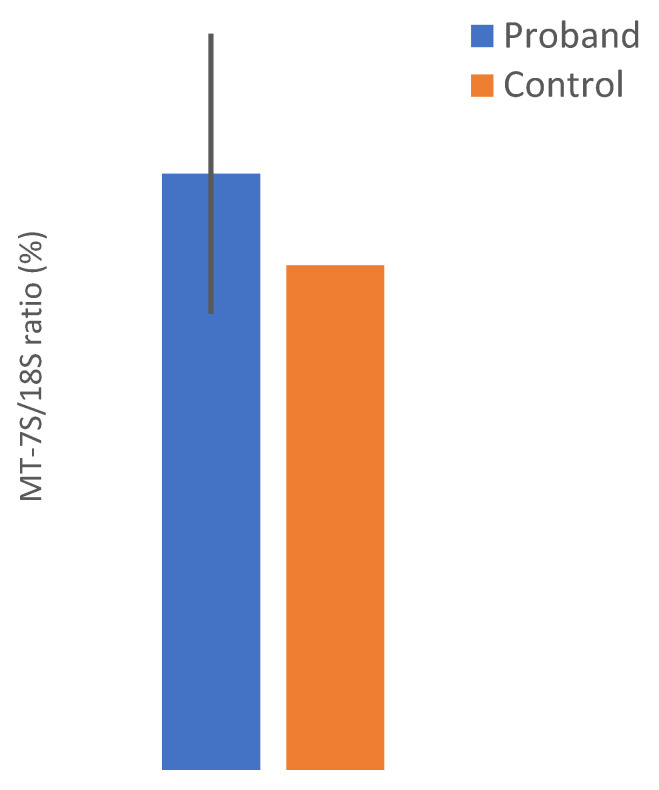
Relative mtDNA content (MT-7S/18S) in muscle of the proband’s mother compared to age-matched controls.

**Figure 5 life-13-00554-f005:**
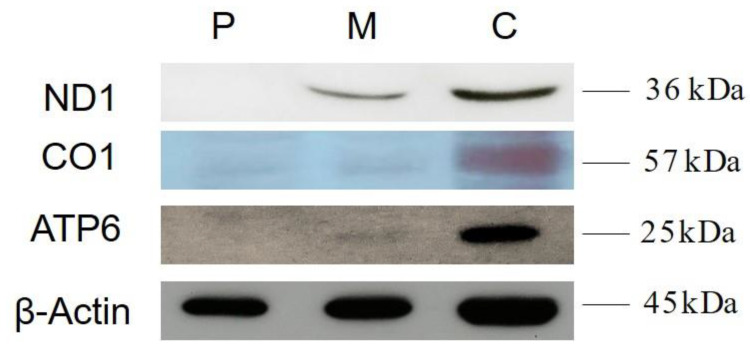
Western blot analysis of mitochondrial proteins shows a severely decreased amount of all four respiratory complex subunits in patient muscle and partially in her mother. P = Patient, M = Mother and C = Control.

**Table 1 life-13-00554-t001:** Clinical phenotypes and the genetic characteristics of the cases reported from literature.

Locus	Mutation	Homoplasmy	Heteroplasmy	Status	MitoTIP	Disease	First Report
MT-COI/ MT-TS1 precursor	m.7443A>G	+	-	Reported	-	Hearing loss	Pandya et al., 1999 [13]
MT-COI/ MT-TS1 precursor	m.7444G>A	+	-	Reported	-	Aminoglycoside-induced deafness and non syndromic hearing loss	Zhu et al., 2006 [12]
	m.7444G>A (with m.3460G>A or m.14484T>C)					LHON	Brown et al., 1995 [25]
	m.7444G>A (with m.1555A>G)					Hearing loss	Pandya et al., 1999 [13]
						Aminoglycoside-induced deafness	Yuan et al., 2005 [26]
	m.7444G>A (with m.1494C>T)					Aminoglycoside-induced and non syndromic hearing loss	Yuan et al., 2007 [27]
	m.7444G>A (with m.6498C>A)					Non syndromic hearing loss, diabetes and congenital visual loss	Mkaouar-Rebai et al., 2013 [28]
MT-TS1 precursor	m.7445A>C	+	-	Reported	-	Hearing loss	Pandya et al., 1999 [13]
MT-TS1 precursor	m.7445A>G	+	+	Confirmed	-	Sensorineural hearing loss	Reid et al., 1994 [14]
						Progressive hearing loss and palmoplantar keratoderma	Sevior et al., 1998 [29]
						Sensorineural deafness and NEPPK	Martin et al., 2000 [30]
MT-TS1 precursor	m.7445A>T	+	-	Reported	-	Sensorineural hearing loss	Chen et al., 2008 [31]
MT-TS1	m.7451A>T	-	+	Reported	80.70%	C-PEO, ptosis	Blakely et al., 2013 [32]
MT-TS1	m.7453G>A	+	-	Reported	68.00%	Fatal neonatal lactic acidosis	Gotz et al., 2012 [33]
						Neonatal lactic acidosis, exercise intolerance, mild ID	Riley et al., 2020 [34]
MT-TS1	m.7456A>G	+	-	Unclear	16.00%	Deafness	Jacobs et al., 2005 [35]
MT-TS1	m.7458G>A	-	+	Reported	86.00%	PEO	Souilem et al., 2010 [36]
MT-TS1	m.7462C>T	+	-	Reported	11.20%	Hearing loss	Uehara et al., 2010 [37]
MT-TS1	m.7471del	nd	nd	Reported	4.30%	Maternally inherited hypertension	Yang et al., 2020 [38]
						Deafness	Tang et al., 2015 [39]
MT-TS1	m.7471_7472insC (reported as m.7472insC)	+	+	Confirmed	-	Hearing loss, ataxia, dysarthria and, occasionally, peripheral sensory neuropathy and focal myoclonus	Tiranti et al., 1995 [40]
						Sensorineural hearing loss, myoclonic epilepsy, ataxia, MR	Jaksch et al., 1998 [19]
						Epilepsia partialis continua, ataxia, lactic acidosis, myopathy, sensorineural hearing loss, severe headaches, and MR	Schuelke et al., 1998 [20]
						Non syndromic sensorineural hearing loss and monomelic amyotrophic	Fetoni et al., 2004 [41]
						Non syndromic sensorineural hearing loss	Hutchin et al., 2001 [42]
MT-TS1	m.7472A>C (with m.7471_7472insC)	+	+	Reported	3.2%	Early onset myopathy and execise intollerance	Pulkes et al., 2005 [43]
						Bilateral hearing loss, MR, fatal neurodegeneration with cognitive decline, epilepsia partialis continua, myopathy, lactic acidosis and ataxia	Cardaioli et al., 2006 [44]
MT-TS1	m.7474A>G	nd	nd	Reported	0.00%	Hearing loss	Zheng et al., 2020 [45]
MT-TS1	m.7474del	nd	nd	Reported	34.80%	Hearing loss and epilepsy	Zhao et al., 2008 [18]
MT-TS1	m.7480T>C	-	+	Reported	46.60%	Progressive mitochondrial myopathy, deafness, dementia and ataxia	Bidooki et al., 2004 [46]
MT-TS1	m.7486G>A	-	+	Reported	50.50%	C-PEO	Bacalhau et al., 2018 [47]
MT-TS1	m.7492C>T	+	-	Reported	0.10%	Hypertension	Liu et al., 2014 [22]
						Hearing loss	Peng et al., 2020 [48]
						Polycystic ovary syndrome-insulin resistance	Dyng et al., 2017 [24]
MT-TS1	m.7496T>C	nd	nd	Reported	58.30%	Hearing Loss	Tang et al., 2015 [39]
MT-TS1	m.7497G>A	+	+	Confirmed	Pathogenic	Severe progressive myopathy, muscle weakness and increase exercise intolerance	Jaksch et al., 1998 [7]
						Exercise intolerance, muscle pain and lactic acidemia	Grafakou et al., 2003 [49]
						Muscular weakness, atrophy and severe dystrophic myopathy	Muller et al., 2005 [50]
MT-TS1	m.7501T>A	nd	nd	Reported	1.90%	Cardiovascular disease	Zaragoza et al., 2010 [21]
						Renal disease patient	Imasawa et al., 2014 [23]
MT-TS1	m.7502C>T	nd	nd	Reported	8.20%	Tic disorder	Jiang et al., 2020 [51]
MT-TS1	m.7505T>C	+	-	Reported	58.60%	Maternally inherited hearing loss	Tang et al., 2010 [52]
MT-TS1	m.7506G>A	-	+	Reported	81.40%	PEO and hearing loss	Cardaioli et al., 2007 [17]
MT-TS1	m.7507A>G	+	-	Reported	-	Cardio-respiratory failure and fatal lactic acidosis, severe hearing loss and progressive exercise intolerance	McCann et al., 2015 [53]
MT-TS1	m.7510T>C	-	+	Confirmed	Pathogenic	Non syndromic sensorineural hearing loss	Hutchin et al., 2000 [15]
MT-TS1	m.7511T>C	+	+	Confirmed	Pathogenic	Non syndromic hearing loss	Sue et al., 1999 [16]
MT-TS1	m.7512T>C	+	+	Reported	64.20%	MERRF/MELAS overlap syndrome	Nakamura et al., 1995 [54]
						Sensorineural hearing loss, myoclonic epilepsy, ataxia, MR	Jaksch et al., 1998 [19]
						Sensorineural hearing loss, myoclonus epilepsy, ataxia, severe psychomotor retardation, short stature, and diabetes mellitus	Ramelli et al., 2006 [55]
						MELAS syndrome	Lindberg et al., 2008 [56]

Legend—LHON (Leber hereditary optic neuropathy); NEPPK (non-epidermolytic palmoplantar keratoderma); C-PEO (chronic-progressive external ophthalmoplegia), ID (intellectual disability); MR (mental retardation), MERRF (myoclonic epilepsy with ragged-red fibers); MELAS (myopathy with encephalopathy, lactic acidosis and stroke-like episodes); nd (not determined).

## Data Availability

Data are available from the corresponding author upon a reasonable request.

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
