# Peer review of "The Mitochondrial tRNASer(UCN) Gene: A Novel m.7484A>G Mutation Associated with Mitochondrial Encephalomyopathy and Literature Review"

_life, 2023, doi:10.3390/life13020554_

Round 1
Reviewer 1 Report
Borgione et al., first report a case with a novel m.7484A>G mutation associated with mitochondrial encephalomyopathy. They also tried to explore the role of mutation with morphologic and biochemical analysis. Finally, they made a conclusion about the reported cases of clinical phenotypes and the genetic characteristics from literature. The quality and the amount of data are satisfactory for the publication in life. However, several points need to be addressed.
1. The images in Figure 2 is very vague, the author show include images with higher resolution.
2. The authors mentioned he mtDNA copy number analysis showed normal mtDNA content in the mother’s muscle when compared to tissue and age-matched pooled controls. But they didn’t show the results.
3. Figure 4: The loading control of the samples is not consistent, it is necessary to make quantification with three independent experiment.
In the result section, the subheading is present with methods, it is better to present with results.
Minor points:
1) The original figure of western blot should be displayed.
2) Figure 2: scale bar should be included.
3) Figure 4: the molecular weight should be added.
Author Response
Dear Editors and Reviewers
Thanks for your comments. We would like to submit our revised version of the manuscript for possible publication in your journal.
Reviewer 1
Borgione et al., first report a case with a novel m.7484A>G mutation associated with mitochondrial encephalomyopathy. They also tried to explore the role of mutation with morphologic and biochemical analysis. Finally, they made a conclusion about the reported cases of clinical phenotypes and the genetic characteristics from literature. The quality and the amount of data are satisfactory for the publication in life. However, several points need to be addressed.
- The images in Figure 2 is very vague, the author show include images with higher resolution.
R: Figure 2 has been changed including images with higher resolution
- The authors mentioned he mtDNA copy number analysis showed normal mtDNA content in the mother’s muscle when compared to tissue and age-matched pooled controls. But they didn’t show the results.
R: We added a new Figure 4 showing the results of qRT-PCR
- Figure 4: The loading control of the samples is not consistent; it is necessary to make quantification with three independent experiments.
R: Unfortunately, not enough biological tissue was available to repeat other experiments
In the result section, the subheading is present with methods, it is better to present with results.
R: We removed the subheading in the result section
Minor points:
- The original figure of western blot should be displayed.
R: we added the available original figures of western blot on “supplementary file for review use only”.
2) Figure 2: scale bar should be included.
R: scale bar was included in Figure 2
3) Figure 4: the molecular weight should be added.
R: the molecular weight was added to Figure 4 now renamed Figure 5
Hoping in positive feedback we look forward to hearing from you soon.
Kind regards,
Vincenzo Di Stefano
Department of Biomedicine, Neuroscience and advanced Diagnostic (BIND), University of Palermo, Italy.
Reviewer 2 Report
The manuscript by Borgione et al is a case report and review of the literature about mutations in mitochondria tRNA gene. The manuscript is constructed well and the topic is clinically relevant in rare cases.
Authors also reviewed articles which described mutations which observed in patients with mitochondria deficiencies. In their discussion part, authors shows clinical phenotype in the table is attractive to clinicians and patients with mitochondria diseases.
Author Response
Dear Editors and Reviewers
Thanks for your comments. We would like to submit our revised version of the manuscript for possible publication in your journal.
Reviewer 2
The manuscript by Borgione et al is a case report and review of the literature about mutations in mitochondria tRNA gene. The manuscript is constructed well, and the topic is clinically relevant in rare cases.
Authors also reviewed articles which described mutations which observed in patients with mitochondria deficiencies. In their discussion part, authors show clinical phenotype in the table is attractive to clinicians and patients with mitochondria diseases.
R: Thank you for your appreciation and precious considerations.
Hoping in positive feedback we look forward to hearing from you soon.
Kind regards,
Vincenzo Di Stefano
Department of Biomedicine, Neuroscience and advanced Diagnostic (BIND), University of Palermo, Italy.
Round 2
Reviewer 1 Report
The quality and the amount of data are satisfactory for the publication in life.